# Comparison of PB1-F2 Proximity Interactomes Reveals Functional Differences between a Human and an Avian Influenza Virus

**DOI:** 10.3390/v15020328

**Published:** 2023-01-24

**Authors:** Joëlle Mettier, Clémentine Prompt, Elise Bruder, Bruno Da Costa, Christophe Chevalier, Ronan Le Goffic

**Affiliations:** INRAE, UVSQ, UMR892 VIM, Université Paris-Saclay, 78350 Jouy-en-Josas, France

**Keywords:** influenza virus, PB1-F2, BioID2, interactome, virulence factor, 14-3-3 proteins, host response

## Abstract

Most influenza viruses express the PB1-F2 protein which is regarded as a virulence factor. However, PB1-F2 behaves differently in avian and mammalian hosts, suggesting that this protein may be involved in the species barrier crossings regularly observed in influenza viruses. To better understand the functions associated with this viral protein, we decided to compare the BioID2-derived proximity interactome of a human PB1-F2 from an H3N2 virus with that of an avian PB1-F2 from an H7N1 strain. The results obtained reveal that the two proteins share only a few interactors and thus common functions. The human virus protein is mainly involved in signaling by Rho GTPases while the avian virus protein is mainly involved in ribonucleoprotein complex biogenesis. PB1-F2 H3N2 interactors include several members of the 14-3-3 protein family, a family of regulatory proteins involved in many signaling pathways. We then validated the interaction with 14-3-3 proteins and were able to show that the association of H3N2-PB1-F2 with YWHAH increased the activity of the antiviral sensor MDA5, while H7N1-PB1-F2 had no effect. Collectively, these results show that PB1-F2 can associate with a large range of protein complexes and exert a wide variety of functions. Furthermore, PB1-F2 interactome differs according to the avian or human origin of the protein.

## 1. Introduction

Influenza A viruses (IAV) are a threat to human and veterinary health. They belong to the *Orthomyxoviridae* family and their genome consists of segmented negative-polarity single-stranded RNA. The error-prone replication process of IAV, as well as the segmented nature of their genome, give them remarkable host adaptation abilities. As a result, species barrier crossings are commonly observed and human infections with avian viruses are regularly described [1]. Human infections with avian IAV can be particularly dramatic as the population lacks immunity against these emerging viruses. In recent years, human infections with H5N1 and H7N9 strains have been particularly virulent, with mortality rates as high as 66% and 40%, respectively [2]. Therefore, determining the factors enabling or facilitating these species barrier crossings is an important challenge in the field of influenza virus research.

The best known of these factors are the sialic acids constituting the influenza virus receptors. The hemagglutinin (HA) glycoprotein of avian influenza viruses binds preferentially to sialic acids attached to a galactose residue via an α2,3 linkage. In contrast, HA proteins of human influenza viruses preferentially bind α2,6-linked sialic acids [3]. This difference is a major obstacle to crossing the species barrier. Another host factor implicated in host range restriction is ANP32. This host protein mediates the formation of dimers of polymerase complex required for genome replication [4]. Chicken ANP32A supports virus replication in avian cells and its expression in human cells has been shown to enhance avian virus replication [5]. Acquisition of host-adapting mutations in avian strains, such as PB2 E627K, enables polymerase activity to be supported by the shorter ANP32 proteins typical of mammalian hosts [6]. This example is representative of the crossing of species barriers that the influenza virus can achieve. However, several studies showed that emerging avian viruses can bypass this barrier and infect mammalian hosts without prior adaptation to sialic acids or acquisition of mutations in polymerase [7]. This highlights the contribution of other viral factors in species barrier crossings.

The PB1-F2 protein is a virulence factor expressed by some influenza viruses [8,9]. It is encoded by an alternative open reading frame located on genomic segment #2. Epidemiological studies revealed that PB1-F2 expression is not under the same regulation processes in mammal and avian hosts. Although most avian viruses express a full-length PB1-F2, the majority of human viruses express a non-functional truncated protein (i.e., less than 79 aa) [10,11], particularly H1N1 strains. Indeed, since 1947, almost all isolated human seasonal H1N1 strains have encoded a truncated 57 aa-long form of PB1-F2. This also included the last 2009 pandemic A/H1N1 virus, which expressed only the first 11 amino acids of PB1-F2 [12]. This evolutionary loss over time suggests that the loss of expression of PB1-F2 is beneficial for the fitness of H1N1 isolates in humans and more generally in mammals. Overall, PB1-F2 expression seems to be positively selected in avian hosts and conversely in mammals. PB1-F2 is thus suspected to play a role in the crossing of the species barrier.

These phylogenetic observations are supported by studies on the link between PB1-F2 and virulence. First, all pandemic viruses of the last century expressed a functional PB1-F2 that was of avian origin [13,14]. The direct contribution of PB1-F2 to virulence was demonstrated for the H1N1 virus that caused the 1918 influenza pandemic [15]. Furthermore, seasonal H1N1 viruses cause milder symptoms than pre-40s H1N1 seasonal strains and contemporary H3N2 seasonal strains [16,17] that express a full-length PB1-F2 protein. This suggests a link between PB1-F2 and IAV virulence in humans. The role of PB1-F2 proteins from avian H5N1 viruses has been studied in mice and their expression has been associated with immuno-pathological processes [18,19,20,21,22,23]. In the avian host, PB1-F2 has been shown to contribute to an optimized spread of the virus without increasing the virulence in chickens [22,24]. These results are consistent with the positive selection discussed above. However, PB1-F2 of HPAI viruses was shown to be a pathogenicity determinant in duck [23]. Thus, even if several studies pointed out its role in pathogenicity in mammals, the function of PB1-F2 has not yet been fully understood, especially within the avian host in which the high conservation of PB1-F2 suggests an essential role [25].

We recently investigated the contribution of the accessory protein PB1-F2 to species specificity. For this purpose, we produced human chimeric viruses in which segment #2, encoding PB1-F2, was of avian origin [26]. Unexpectedly, by knocking-out this protein within the chimeric virus, we made a virus more inflammatory than the virus able to express PB1-F2. This result goes against our previous results and illustrates the complexity of the function of PB1-F2 which seems to depend on its protein environment. To clarify these apparent contradictory data, we undertook a comparison of the PB1-F2 protein interactomes from an avian H7N1 virus and a human H3N2 virus. The differential interactomic landscapes of the two proteins indicated common and differential pathways targeted by PB1-F2, including pathways involved in the host response that could explain the differences observed with chimeric viruses.

## 2. Materials and Methods

### 2.1. Purification of Recombinant PB1-F2 and Fibrillation Assays

PB1-F2 protein of avian influenza A/Turkey/Italy/977/1999 [H7N1], human influenza A/Scotland/20/1974 [H3N2] and human influenza A/WSN/1933 [H1N1] viruses were expressed and purified as described previously [27,28]. Briefly, the gene encoding either full-length PB1-F2(1–90) protein were cloned into the pET22b+ expression vector (Novagen, Fontenay sous Bois, France) to express His6-tagged protein versions. Transformed competent BL-21 Rosetta cells (Stratagene, San Diego, CA, USA) were incubated with 1 mM isopropyl 1-thio-B-D-galac-topyranoside for 4 h at 37 °C. After cell lysis and solubilization in 8 M urea buffer, the recombinant PB1-F2-His proteins were purified from inclusion bodies on a Hitrap-IMAC column using the AKTA Purifier-100 FPLC chromatography system (GE Healthcare, Buc, France). Fractions collected containing PB1-F2-His proteins were further purified by size exclusion chromatography on a 120 mL Superdex 200 column. Urea was removed on a G25 desalting column equilibrated with 5 mM ammonium acetate buffer (pH 5). PB1-F2-His proteins were lyophilized and stored at −20 °C. Lyophilized protein powder was dissolved in 5 mM sodium acetate pH 5 buffer and its concentration was determined by measuring optical density at 280 nm using extinction coefficient deduced from its composition of 28,990 M^-1^cm^-1^. PB1-F2 fibers were produced as previously described [29]. PB1-F2 was incubated with 0.01% (*w*/*v*) SDS in 5 mM sodium acetate buffer (pH 5) for 45 min at room temperature.

### 2.2. Thioflavin T Fluorescence Measurements

Thioflavin T (ThT) binding assays were performed by adding 0.01% SDS and 25 µM of freshly prepared ThT, to 5 µM of PB1-F2-HisTag samples, all in in 5 mM sodium acetate buffer, pH 5. The fluorescence measurements were performed after 30 min of incubation at room temperature on a Tecan Infinite M200 PRO plate reader (λexcitation = 435 nm, λemission = [450; 600] nm).

### 2.3. Electronic Microscopy

Electron micrographs were acquired using a transmission electron microscope at 80-kV excitation voltage as previously described [28].

### 2.4. Cell Culture

Cell lines 293T and A549 were obtained from the ATCC (American Type Culture Collection). Cells were grown in DMEM medium at 37 °C and incubated in a regulated atmosphere of 5% CO_2_. The culture media used were supplemented with 10% SVF, 2 mM L-glutamine, 100 units/mL penicillin, and 100 μg/mL streptomycin.

### 2.5. BioID2 Pull Down

The two PB1-F2 and the GFP ORFs were cloned into a pcDNA3.1 BioID2-HA vector (MCS-BioID2-HA was a gift from Kyle Roux (Addgene plasmid # 74224; http://n2t.net/addgene:74224, accessed on January 2023; RRID: Addgene_74224)) [30]. For large scale BioID2 pull down, 13 × 10^6^ A549 cells were plated in a T-175 flask. The cells were transfected with FuGENE^®^ HD reagent (Promega, Charbonnières-les-Bains, France; #E2312) using a 4/1 reagent/DNA ratio following manufacturer recommendations. Twenty-four hours post-transfection, the cells were incubated with 50 µM D-biotin for 24 h and then trypsinized. After two PBS washes, the cells were lysed in 3.5 mL of lysis buffer (150 mM NaCl; 0.1% SDS; 1% Igepal; 0.5% sodium deoxycholate; 1 mM EDTA; 25 mM Tris pH 7.4; 1 mM DTT and a complete protease inhibitor cocktail (Roche, #4693159001). After 10 min incubation at 4 °C, lysates were centrifuged at 16,000× *g* for 30 min at 4 °C. Collected supernatants were filtered through an ultrafiltration membrane with a nominal molecular weight limit of 3 kDa (MERCK #UFC8003) at 5000 g for 45 min at 4 °C. The filtered supernatants were incubated with 250 μL of streptavidin-coupled magnetic beads (InvitrogenTM, #11205D) overnight at 4 °C under gentle agitation. Beads were previously equilibrated by 4 successive washes: 3 with PBS and 1 with lysis buffer. After purification of the biotinylated proteins, beads were collected using a magnetic stand and washed once with 2% SDS, once with wash buffer containing 500 mM NaCl, 1 mM EDTA, 1% Triton X-100, 50 mM HEPES and 0.1% sodium deoxycholate and once with wash buffer containing 300 mM NaCl; 1 mM EDTA; 0.5% sodium deoxycholate; 0.5% Igepal and 10 mM Tris pH 8. Finally, the beads were washed three times with 50 mM Tris pH 7.4. To elute the bound proteins, the beads were resuspended in 30 μL of Laemmli buffer (25 mM Tris-HCl; 192 mM glycine; 10% SDS; 25% glycerol and 0.01% bromophenol blue) and incubated for 15 min at 95 °C. The eluates were separated from the beads and stored at −20°C. Resuspended proteins were finally either subjected to Western blot analysis or digested with trypsin and then analyzed via mass spectrometry. We used a GFP-BioID2 vector as a control for non-specific biotinylation. Thus, to identify specific interactors, we first excluded from the PB1-F2 datasets all proteins also precipitated from cells transfected with the GFP-BioID2 vector. The remaining proteins were filtered against the CRAPome database [31].

### 2.6. Immunofluorescence

A549 cells were plated on coverslips. The cells were transfected and incubated with D-biotin as described in the “BioID2 pull-down” section. The cells were fixed with 4% paraformaldehyde, permeabilized with 0.1% Triton X-100 and finally incubated in 0.1M glycine. Then, cells were blocked with 3% bovine serum albumin and 0.05% Tween 20 and incubated with the primary antibodies of interest for 1 h at room temperature: rabbit anti-HA (Sigma Aldrich, Saint-Quentin-Fallavier, France; #H6908; dilution 1/100) or Alexa Fluor^®^ 594 streptavidin (Molecular Probes^TM^, Eugene, OR, USA; #10626153; dilution 1/500). Cells were washed twice with PBS and incubated with anti-rabbit Alexa Fluor^®^ 488 (abcam #ab150077; dilution 1/2000). Finally, the cells were washed 3 times with PBS before mounting the slides using a medium containing DAPI (Molecular Probes, #15247528). Imaging was performed with an inverted Zeiss Axio Observer Z1 microscope. The images were acquired and processed using ZEN lite software (Zeiss, Oberkochen, Germany).

### 2.7. Western Blot Analysis

After cells lysis samples were loaded and separated on a 12.5% polyacrylamide gel. Gels were transferred (semi-dry, 2.5 A/25 V/3 min) onto Immobilon-P nitrocellulose membranes with a 0.2 µm pore size (Millipore, Molsheim, France). Membranes were blocked with PBS containing 2% BSA and 0.2% Triton X-100. BioID2 constructs expression were confirmed by Western blot analysis using a rabbit anti-HA (Sigma #H6908; dilution: 1/100) and anti-rabbit HRP (KPL #5450-0010: dilution 1/5000) antibodies. Biotinylated proteins were visualized using HRP-conjugated streptavidin (abcam #ab7403, dilution 1/5000) overnight at 4 °C in 0.2% Triton X-100 PBS.

### 2.8. Coimmunoprecipitation Assays

293T cells (12-well plate at 3.5 × 10^5^ cells/well) were cotransfected with pCI-H3N2-PB1-F2 or pCI-H7N1-PB1-F2 and pcDNA3.1 encoding a HA-tagged 14.3.3 protein (YWHAB, YWHAE, YWHAH or YWHAS) for 24 h. Transfected cells were then lysed for 30 min at 4 °C in 100 µL of ice-cold lysis buffer (50 mM Tris-HCl [pH 7.4], 2 mM EDTA, 150 mM NaCl, 0.5% NP-40) and a complete protease inhibitor cocktail (Roche, Basel, Switzerland; #4693159001). Coimmunoprecipitation experiments were performed on cytosolic extracts. Cell lysates were incubated for 4 h at 4 °C with 2 µL of anti-Flag antibody (Sigma, #F4042) coupled to 40 µL of agarose beads (Protein G Sepharose 4 Fast Flow, #17-0618-01, GE Healthcare). The beads were then washed 3 times with lysis buffer and 1 time with PBS, and proteins were eluted in 40 µL of Laemmli buffer at 95 °C for 5 min and then subjected to SDS-PAGE and immunoblotting.

## 3. Results

### 3.1. Fibrillation Capacities of PB1-F2

In a previous study, we showed that PB1-F2 from human H3N2 and avian H7N1 viruses had different pro-inflammatory capabilities. In mouse infections, the avian PB1-F2 was much more inflammatory than the human PB1-F2. However, a chimeric H3N2 virus expressing H7N1 PB1-F2 had reduced pro-inflammatory activity compared to wild-type H3N2 virus [26]. These observed differences could be explained by more or less pronounced fiber-forming abilities. As a matter of fact, PB1-F2 can fibrillate within hydrophobic structures and therefore generate a cytopathic effect [28]. In addition, there is a correlation between fiber formation and the pro-inflammatory properties of PB1-F2 [27]. We therefore hypothesized that the two different effects on the inflammatory response of the PB1-F2 studied might be related to their ability to form fibers. Using recombinant proteins, we induced in vitro fibrillation of PB1-F2 H3N2 and H7N1. We also used PB1-F2 from A/WSN/1933 (H1N1) virus as a positive control. As shown in Figure 1A, we observed a fluorescence peak of ThT at approximately 490 nm after excitation at 435 nm in the presence of the three proteins, which is characteristic of amyloid-like fibers [32]. 

In the presence of fibers formed by PB1-F2 from H3N2 and H7N1 viruses, the ThT fluorescence signal peaked at a wavelength between 494 and 500 nm, comparable to that of the H1N1 PB1-F2 control. In addition, observation of the fibers by transmission electron microscopy (Figure 1B,C) revealed a very similar structural conformation between the proteins from the two different subtypes. Since the fibrillation capacities of the two PB1-F2 are comparable, the previously observed difference in pro-inflammatory activity is not due to this factor.

### 3.2. Expression of BioID2-Tagged PB1-F2

Since the ability of the PB1-F2 studied to fibrillate cannot explain the different effects on the inflammatory response, we then hypothesized that these two proteins must establish distinct protein/protein interactions (PPIs) within the infected host to induce different effects. We therefore investigated the interactomic profiles of the two virulence factors using a proximity-dependent biotin labeling approach. To this end, we fused the two PB1-F2 to the N-terminal part of a BioID2 tag, a small biotin ligase derived from the *Aquifex aeolicus* [30]. We also used a vector allowing the expression of a BioID2-tagged GFP protein to identify non-specific PPIs. As shown in Figure 2A, H3N2-PB1-F2-BioID2, H7N1 PB1-F2-BioID2 and GFP-BioID2 constructs were correctly expressed in A549 cells after transfection. Addition of biotin to the cell medium revealed a smear of labeled proteins, attesting to the functionality of the BioID2 tag (Figure 2B). It should be noted that when biotin is not added to the medium, Western blots show no signal, demonstrating the specificity of the signal (data not shown). The cellular localization of the fusion proteins was also verified and the result was consistent with what is described for the different PB1-F2s studied: a cytoplasmic punctate localization as well as a less marked nuclear signal [33,34]. Biotin labeling also showed that biotinylated proteins colocalized with BioID2-tagged proteins (Figure 2C).

### 3.3. PB1-F2 Proximity Interactomes Comparison

After controlling the expression, functionality and localization of the fusion proteins, we set out to identify the host proteins that interact with the two PB1-F2 studied. We performed a BioID2 screen using LC-MS/MS analysis, to compare H3N2 and H7N1 PB1-F2 interactomes. We used the GFP-BioID2 vector as a control for non-specific biotinylation, we therefore excluded all proteins identified with the GFP-BioID2 bait. Hence, we identified a total of 234 proteins interacting with the two PB1-F2 studied but only 145 proteins were considered potentially specific to PB1-F2s (Figure 3A). Then, we confronted “PB1-F2-specific” proteins with the CRAPome database (https://reprint-apms.org/, accessed on January 2023; [31]), which helped us determine which polypeptides represent *bona fide* interactors vs. those that are background contaminants. We opted to exclude from the analysis proteins with a distribution frequency higher than 50% in the human “Proximity-Dependent Biotinylation” database. This allowed us to exclude likely contaminants and reduce the number of interactors to 50 and 45 for PB1-F2 from H7N1 and H3N2, respectively. The identity of these interactors is detailed in Appendix A. To visualize if a cell compartment is specifically targeted by the two PB1-F2, we used the “SubcellulaRVis” application (http://phenome.manchester.ac.uk/subcellular/, accessed on January 2023; [35]). This application is used to determine the enrichment of subcellular locations of gene lists. Consistent with the high cytoplasmic expression observed for both proteins by immunocytochemistry (Figure 2C), the cytoplasmic compartment is enriched into interactors for both PB1-F2 (FDR < 0.01). However, we also observed differences between the two PB1-F2: H3N2 PB1-F2 additionally interacts with elements exported to the extracellular compartment or localized in intracellular vesicles, whereas PB1-F2 from H7N1 interacts with proteins addressed to the nucleus (Figure 3B). These differences in cellular localization of the PB1-F2 partners suggest different modes of action.

### 3.4. PB1-F2 Network Analysis

Among the PB1-F2 interactors, after CRAPome filtering, only 10 of them interact with both proteins (Figure 4A). This wide diversity of interactors between the two proteins is consistent with their opposite functions observed in the mouse model [26]. Using Gene Ontology (GO), we analyzed the functions associated with these 10 proteins. A single GO-term was associated with this set of genes: “Basolateral plasma membrane” (FDR: 8.2 × 10^−6^), confirming the affinity of PB1-F2 for membranes [27,28,36].

To go further in this approach and give a functional meaning to the two protein–protein interaction networks, we used the online interface “Metascape” (https://metascape.org/, accessed on January 2023) [37]. The separate ontological analysis of each network of interactors allowed the identification of common functions potentially regulated by both PB1-F2, but also functions specifically modulated by a particular PB1-F2. Figure 4B details the enriched ontology clusters identified by Metascape. Among the functions associated with both PB1-F2, we observe that some interactors are involved in several functions or signaling pathways. Indeed, for instance, EGFR, the receptor of the epidermal growth factor cytokine is implicated in 3 out of 6 pathways identified as commonly regulated by the 2 PB1-F2 (“Vesicle-mediated transport”, “viral life cycle” and “cell morphogenesis”). Moreover, ITGB1, an integrin protein interacting with the H3N2 PB1-F2, is also involved in 4 signaling pathways in which it seems to play a central role (“Signaling by Rho GTPases”, “viral life cycle”, “cell morphogenesis”, and “VEGFA-VEGFR2 signaling pathway”). Of note, some of the identified functionalities, such as “skin development” or “Arrhythmogenic right ventricular cardiomyopathy” must be considered to be not relevant, since Metascape cannot restrict analysis to a specific organ (i.e., respiratory tract in our case).

The interaction networks have been modeled in Figure 5 and show the high connectivity of the different members interacting with PB1-F2. H3N2 PB1-F2 interacting network is represented in Figure 5A. Proteins colored in red are involved in the following signaling pathways: Signaling by Rho GTPases (R-HSA-194315); Signaling by Rho GTPases, Miro GTPases and RHOBTB3 (R-HSA-9716542); PID A6B1 A6B4 Integrin Pathway (M239). The H7N1 PB1-F2 interacting network is shown in Figure 5B, red nodes representing protein implicated in the following GO-terms: ribonucleoprotein complex biogenesis (GO:0022613); ribonucleoprotein complex assembly (GO:0022618); ribonucleoprotein complex subunit organization (GO:0071826). Thus, if we exclude the functions conserved by both PB1-F2 and focus on the specificities of each PB1-F2, the striking information is the very large functional differences exerted by the specific interactors of each protein. Indeed, while the H3N2 PB1-F2 appears implicated in membrane-associated processes such as transport or signal transduction, the PB1-F2 interaction network of the H7N1 virus is strongly involved in RNA metabolism, “ribonucleoprotein complex assembly” being a major component of this signature. Based on this ontological analysis, we can therefore assume that within the infected cell, the PB1-F2 of the human virus will mainly act at the level of signal transduction, whereas that of the avian virus will have an impact on the RNA transcription processes. 

### 3.5. H3N2 PB1-F2 Interact with 14-3-3 Proteins to Modulate Host Response

The protein–protein interaction network specific to PB1-F2 of the H3N2 strain shows a strong involvement of proteins of the “14-3-3” family (Figure 5A dotted red ellipse). Indeed, among the proteins found in the proximity interactome of H3N2 PB1-F2 are three members of the 14-3-3 protein family: YWHAG (14-3-3 protein γ), YWHAH (14-3-3 protein η) and SFN (14-3-3 protein σ). We therefore studied these interactions in more detail. We first validated the association of proteins by a pull-down experiment. We were not successful in cloning or obtaining an expression plasmid encoding YWHAG; however, we were able to co-express YWHAS, YWHAB, YWHAE and YWHAH with PB1-F2 of the H3N2 strain. As shown in Figure 5C, only YWHAB and YWHAH were co-precipitated together with PB1-F2. YWHAB had also been identified as an H3N2 PB1-F2 interactor in the BioID2 screen, but the interaction was not strong enough to pass the significance thresholds imposed in the analysis using the CRAPome application (data not shown). We then focused on YWHAH to provide a functional insight in this PB1-F2 specific binding. The potential interaction between YWHAH and PB1-F2 from H7N1 was tested and no binding could be seen (Appendix A). YWHAH was recently shown to promote antiviral innate immunity through binding to MDA5 [38]. We therefore set up a functional assay to measure interferon (IFN) activity under conditions of MDA5 overexpression in combination with YWHAH and PB1-F2. In this assay, IFN activity is induced by overexpression of MDA5 (Figure 5D). Interestingly, while co-expression of YWHAH and H7N1 PB1-F2 had no effect on MDA5-induced IFN activity, co-expression of H3N2 PB1-F2 exacerbated the IFN activity. The increase in activity is modest (+60%), but statistically significant. It reflects the differences in interactomes demonstrated for the 2 studied PB1-F2 and is in accordance with the 14-3-3 protein binding capacities of H3N2 PB1-F2.

## 4. Discussion

21 years after its discovery, PB1-F2 still remains enigmatic on many levels. Both functionally and structurally, this protein remains complicated to study and subject to debate. The action of PB1-F2 on innate immunity is typically controversial as it is described both for its inhibitory effects and for its ability to exacerbate the host response [18,19,20,39,40,41]. However, depending on the strain considered, the functions associated with PB1-F2 could be radically different, which is what our work tends to show.

In this paper, we compared the interactomes of two PB1-F2 endowed with opposite behavior in vivo. As a matter of fact, in the mouse model and using a chimeric 7:1 H3N2 virus harboring an H7N1-origin segment 2 and therefore expressing the H7N1 PB1-F2, we have shown that the chimeric virus induced a lower inflammatory response compared to the wild-type H3N2 [26]. The fibrillation capacities of these two proteins are very comparable and are therefore probably not the cause of the differences observed in the infected mice. These two proteins, one from an avian virus and the other from a human virus, display 78% homology (Figure 1A). From the results presented in Figure 3, it appears that the interactomic differences are more related to a different localization of the two proteins rather than to a different protein structure. To support these hypotheses, the predictive analysis of the cellular localization of the two proteins by the LocTree3 software [42] indicates an extracellular localization in the human virus strain (Expected Accuracy 81%), while it predicts a nuclear localization in the avian strain (Expected Accuracy 80%). These predictions appear consistent with the location of their partners identified in our proximity labeling: cytoplasmic, intracellular vesicle and extracellular region for the H3N2 protein; and cytoplasmic and nuclear concerning the H7N1 protein.

Among the interactors identified, we focused on members of the 14-3-3 protein family. Three of them were specifically associated with the H3N2 PB1-F2 protein and could thus explain the differences in effect of the two PB1-F2 studied. We validated the interaction of PB1-F2 with two 14-3-3 proteins by co-immunoprecipitation and identified a functional impact on IFN activity which increased. It therefore appears that by interacting with YWHAH, PB1-F2 may exacerbate its signaling effect within the interferon pathway. This mechanism appears counter-intuitive as it is detrimental to the virus but may explain why many human viruses tend to lack PB1-F2 expression. Alternatively, this result could illustrate the ability of YWHAH to recognize certain forms of PB1-F2 as a danger signal. This effect is rather weak, but contributes to establishing a differential activity between the two PB1-F2 strains. Yet, the sum of all the interactions implemented by the two PB1-F2 must be considered to explore the different activities of PB1-F2.

Our results may add confusion to the function associated with PB1-F2. Obviously, associating a single function to this virulence factor is unrealistic. It seems that this protein is a real Swiss army knife able to bind to multiple protein partners. Its intrinsically disordered structure [28] probably provides the flexibility to adapt to a protein environment that differs from host to host. This protein is also capable of associating into fibers with an amyloid nature. This form of PB1-F2 is probably not the one adopted by PB1-F2 in the interactomes we described in this study. Indeed, PB1-F2 multimerizes when it is located in a hydrophobic environment, i.e., if it is embedded in a membrane. It is likely that this fibrillation only occurs when PB1-F2 does not encounter a cellular binding. Consequently, the expression level of this protein during viral infections is an important parameter to know. Indeed, if the protein is overexpressed, the probability that it will associate with membranes, form fibers and destabilize membrane integrity is greater. Thus, the ratio of free form vs. fibrillated form is also a component of interest. The amount of PB1-F2 produced in an infected mouse lung has already been estimated at 26 pmol of PB1-F2 per mg of infected lung tissue (for comparison, NP is estimated at 650 pmol/mg) [20]; however, it would be useful to quantify it within an infected cell, considering the dynamics of expression and protein half-life (estimated to be around 30 min [9]). Finally, the cell type has also to be considered since the protein environment provided by an epithelial cell to PB1-F2 is different from that of a leukocyte. Therefore, its behavior will be different, as has already been demonstrated [36].

In conclusion, the study of PB1-F2 is complex because of its pleiomorphic structure and its different behavior between strain and host. Its multitude of interactors complicates the understanding of its function and the definition of a functional consensus seems unlikely at the present time.

## Figures and Tables

**Figure 1 viruses-15-00328-f001:**
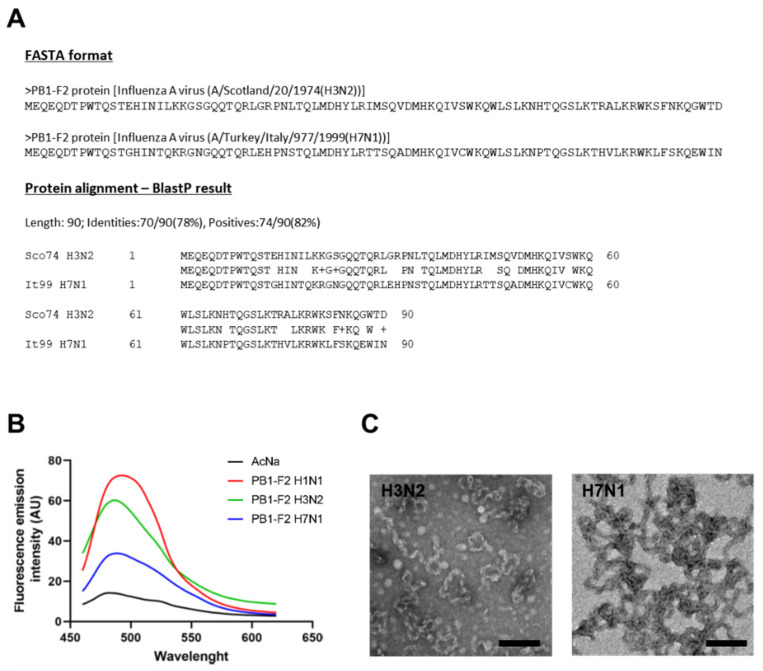
Evaluation of the intrinsic fibrillation capacities of PB1-F2 H3N2 and H7N1. (**A**) Amino acid sequences and protein alignment of the avian and human proteins. (**B**) Measure of the fluorescence increase of ThT after binding with amyloid formed by PB1-F2 H3N2, H7N1 at a final concentration of 10 µM in 5 mM sodium acetate pH5 buffer at 37 °C. PB1-F2 WSN (H1N1) was used as control of fibrillation. AU, absorbance units. (**C**) Observation by electron microscopy of the fibers formed by H3N2 (left) and H7N1 (right) PB1-F2 variants in 10 mM sodium acetate, pH 5 at 37 °C. Scale bars, 1 µm.

**Figure 2 viruses-15-00328-f002:**
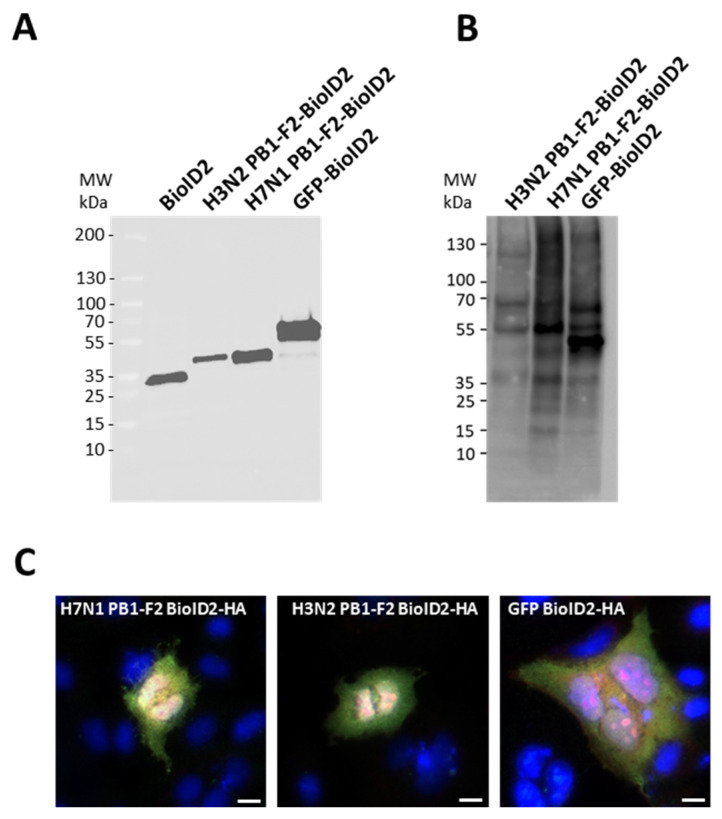
(**A**) Expression of BioID2-HA-tagged constructs were confirmed in A549 cells by SDS-PAGE 24 h post-transfection and analyzed by Western blot using an anti-HA antibody. (**B**) Proteins biotinylated by BioID2 constructs were detected with HRP-conjugated streptavidin. (**C**) A549 cells expressing BioID2-fusion proteins were assessed for localization (green) and proximity biotinylation (red) following the addition of exogenous biotin. Scale bars, 10 µm.

**Figure 3 viruses-15-00328-f003:**
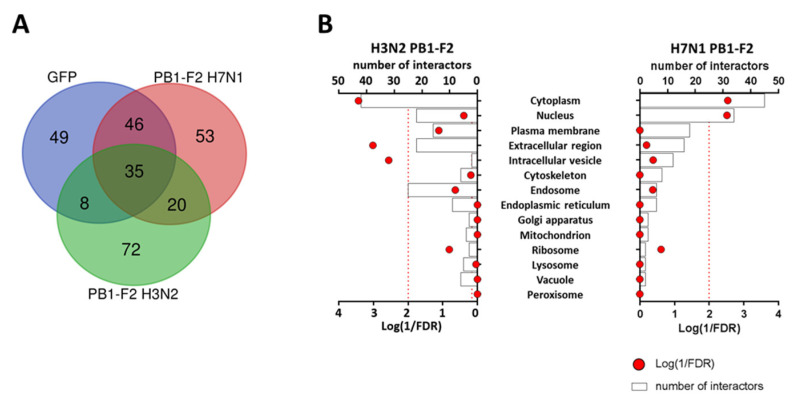
(**A**) Venn diagram representing the repartition of BioID2 fusion proteins interacting partners. (**B**) Details of the compartments targeted by the BioID2 PB1-F2 fusion proteins according to the cellular location of their binding partners.

**Figure 4 viruses-15-00328-f004:**
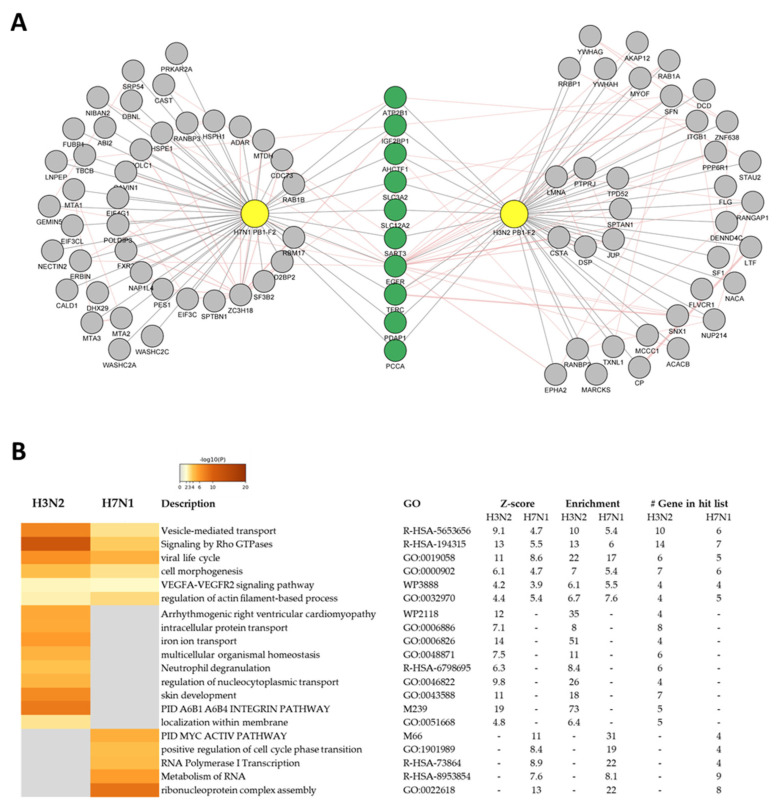
(**A**) Superposition of protein/protein interaction networks of PB1-F2 from H7N1 and H3N2 viruses. The common partners of both PB1-F2 are colored green. (**B**) Cluster of statistically enriched terms (GO/KEGG terms, canonical pathways). Accumulative hypergeometric p-values and enrichment factors were calculated and used for filtering. The remaining significant terms were then hierarchically clustered into a tree based on Kappa-statistical similarities among their gene memberships. Then 0.3 kappa score was applied as the threshold to cast the tree into term clusters. The term with the best p-value within each cluster as its representative term and display them in a dendrogram. The heatmap cells are colored by their *p*-values, gray cells indicate the lack of enrichment for that term in the corresponding gene list.

**Figure 5 viruses-15-00328-f005:**
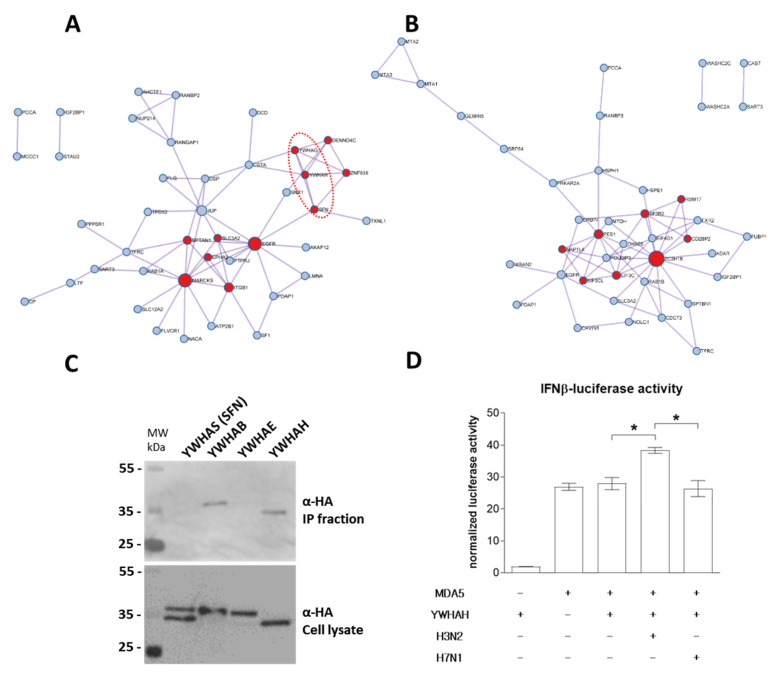
Metascape analysis of PB1-F2 interacting networks. Functional protein–protein interaction network of H3N2 (**A**) and H7N1 PB1-F2 (**B**) are represented. Red nodes represent proteins implicated in “Signaling by Rho GTPases” and in “ribonucleoprotein complex biogenesis” in H3N2 (**A**) and H7N1 (**B**) networks, respectively. The proteins surrounded by a dotted red ellipse are members of the 14-3-3 protein family (**A**). (**C**) Western blot analysis of the PB1-F2–14-3-3 proteins interaction after immunoprecipitation assay. 293T cells were transiently transfected with constructs allowing the expression of HA-tagged 14-3-3 proteins YWHAS, YWHAB, YWHAE and YWHAH together with H3N2 PB1-F2. Immunoprecipitations (IP) were performed with an anti-PB1-F2 antibody. (**D**) 293T cells were transfected with a plasmid encoding an IFNβ-luciferase reporter construct together with MDA5, YWHAH and H3N2 PB1-F2 or H7N1 PB1-F2 expressing plasmids. Twenty-four hours after transfection, cells were lysed and assayed for luciferase activity. * : *p*-value < 0.05.

## Data Availability

Data supporting reported results can be provided by RLG.

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
