# Peer review of "Comparison of PB1-F2 Proximity Interactomes Reveals Functional Differences between a Human and an Avian Influenza Virus"

_viruses, 2023, doi:10.3390/v15020328_

Round 1
Reviewer 1 Report
In this manuscript, Joëlle Mettier 's work compared the functional differences of PB1-F2 protein from human (H3N2) and avian (H7N1) influenza virus. Their results revealed that the two proteins share only few interactors and common functions. The PB1-F2 from human influenza virus protein is mainly involved in signaling by Rho GTPases, while the avian virus protein is mainly associated with ribonucleoprotein complex biogenesis. In summary, their results showed that PB1-F2 of influenza virus can associate with a large range of protein complexes, and exert a wide variety of functions. This finding may help clarify the species barrier between human and avian influenza virus. However, the current data are insufficient to support the conclusions of this manuscript based on following reason: (1) the data provided by the authors only indicated that there are some functional differences of PB2-F2 from H3N2 and H7N1 influenza virus, while these differences should be further validated by reverse genetics. (2) Their results indicated that the association of H3N2-PB1-F2 with YWHAH could increase the activity of the antiviral sensor MDA5, while H7N1-PB1-F2 had no effect. However, the detailed mechanism resulting in this difference is unclear. (3) The title is too big: The author did not compare the sequence differences of PB1-F2 protein from different species, but only discussed the differences of PB1-F2 protein between H3N2 and H7N1 strains. Are you sure that these two strains could represent the differences of PB1-F2 protein between all strains from this two species? Collectively, this manuscript has not yet met the acceptance requirements for publication in this journal.
comments:
1. The language is obscure and difficult to understand.
2. The resolution of Figure 1 is too low.
3. The resolution of Figure 5 is too low.
Author Response
Reviewer #1
Comments and Suggestions for Authors
In this manuscript, Joëlle Mettier 's work compared the functional differences of PB1-F2 protein from human (H3N2) and avian (H7N1) influenza virus. Their results revealed that the two proteins share only few interactors and common functions. The PB1-F2 from human influenza virus protein is mainly involved in signaling by Rho GTPases, while the avian virus protein is mainly associated with ribonucleoprotein complex biogenesis. In summary, their results showed that PB1-F2 of influenza virus can associate with a large range of protein complexes, and exert a wide variety of functions. This finding may help clarify the species barrier between human and avian influenza virus. However, the current data are insufficient to support the conclusions of this manuscript based on following reason:
We thank reviewer #1 for his constructive comments to improve our manuscript.
(1) the data provided by the authors only indicated that there are some functional differences of PB2-F2 from H3N2 and H7N1 influenza virus, while these differences should be further validated by reverse genetics.
Reviewer#1 is right in that a validation by swapping the 2 PB1-F2 of each virus would be a nice demonstration. Unfortunately, since the two ORF encoding PB1-F2 and polymerase are overlapping, the reverse genetic validation is not possible to implement.
(2) Their results indicated that the association of H3N2-PB1-F2 with YWHAH could increase the activity of the antiviral sensor MDA5, while H7N1-PB1-F2 had no effect. However, the detailed mechanism resulting in this difference is unclear.
During viral infection, RLRs, including MDA5, play an essential role in initiating the type I interferon signaling pathway and preventing infection or viral replication in host cells. This process is facilitated by the activity of 14-3-3 proteins, as mentioned in the manuscript: “YWHAH was recently shown to promote antiviral innate immunity through binding to MDA5”. The reference #38 is associated to this statement (Lin, J.P.; Fan, Y.K.; Liu, H.M. The 14-3-3eta chaperone protein promotes antiviral innate immunity via facilitating MDA5 oligomerization and intracellular redistribution. PLoS Pathog 2019, 15, e1007582, doi:10.1371/journal.ppat.1007582.). Hence, previous studies have shown that the expression level of 14-3-3 isoforms is regulated during viral infection and that 14-3-3η (YWHAH) is a key component that promotes the activation of MDA5, necessary for innate antiviral immunity. Here, we show that, during influenza virus infection of a mammalian cell, the expression of H3N2 PB1-F2 exacerbates this process by binding to YWHAH whereas H7N1 PB1-F2 does not.
At this stage of our work, we can only speculate on the underlying mechanism that regulates the increase in YWHAH activity in the presence of PB1-F2 of H3N2 virus. The following sentence has been added to the discussion section to make this point clearer: “It therefore appears that by interacting with YWHAH, PB1-F2 may exacerbate its signaling effect within the interferon pathway. This mechanism appears counter-intuitive as it is detrimental to the virus but may explain why many human viruses tend to lack PB1-F2 expression. Alternatively, this result could illustrate the ability of YWHAH to recognize certain forms of PB1-F2 as a danger signal.”
(3) The title is too big: The author did not compare the sequence differences of PB1-F2 protein from different species, but only discussed the differences of PB1-F2 protein between H3N2 and H7N1 strains. Are you sure that these two strains could represent the differences of PB1-F2 protein between all strains from this two species? Collectively, this manuscript has not yet met the acceptance requirements for publication in this journal.
The title has been modified to follow reviewer #1 recommendation’s: “Comparison of PB1-F2 proximity interactomes reveals functional differences between a human and an avian influenza virus.” Concerning the differences of the PB1-F2 protein between all strains of the two species, it is very complicated to find a consensus sequence for PB1-F2 since this protein is very polymorphic. Therefore, our work should be considered as an attempt to explain the complexity of PB1-F2 biology, and not as a global comparison of the two species.
comments:
- The language is obscure and difficult to understand.
- The resolution of Figure 1 is too low.
- The resolution of Figure 5 is too low.
We will be consulting with the editor to resolve these issues.

Reviewer 2 Report
The paper, " Comparison of PB1-F2 proximity interactomes reveals functional differences between human and avian influenza viruses." by Joëlle Mettier et al, describes the functional differences of PB1-F2 between H7N1 A/Turkey/Italy/977/1999 avian influenza strain and H3N2 A/Scotland/20/1974 human influenza strain. In this paper, BioID2 pull-down was performed, and the two PB1-F2 proteins share only a few interactors. When co-expression of PB1-F2 with YWHAH and MDA5, the H3N2 PB1-F2 exacerbated the IFN activity but the H7N1 PB1-F2 could not. In my opinion, the work is interesting and suitable for publishing in Viruses, but I think some revisions need to be introduced before publication.
Major issues:
The BioID2-derived proximity interactome shows a big difference between human and avian strains of PB1-F2, and the authors suggested these differences are related to the PB1-F2 functions. I suggest that another full-length human or avian PB1-F2 could be added to the BioID2 and mess spec assay to prove that the difference between human and avian PB1-F2 was not random.
Minor issues:
1. Line 63: “acid” should be “acids”.
2. Line 81: “Thus, and even if” should be “Thus, even if”.
3. Line 113: “during 45 min at room temperature” should be “for 45 min at room temperature”.
4. The protein alignment of PB1-F2 could be moved to Figure 1 from the supplementary.
Author Response
Reviewer #2
Comments and Suggestions for Authors
The paper, " Comparison of PB1-F2 proximity interactomes reveals functional differences between human and avian influenza viruses." by Joëlle Mettier et al, describes the functional differences of PB1-F2 between H7N1 A/Turkey/Italy/977/1999 avian influenza strain and H3N2 A/Scotland/20/1974 human influenza strain. In this paper, BioID2 pull-down was performed, and the two PB1-F2 proteins share only a few interactors. When co-expression of PB1-F2 with YWHAH and MDA5, the H3N2 PB1-F2 exacerbated the IFN activity but the H7N1 PB1-F2 could not. In my opinion, the work is interesting and suitable for publishing in Viruses, but I think some revisions need to be introduced before publication.
We thank reviewer #2 for his constructive comments to improve our manuscript.
Major issues:
The BioID2-derived proximity interactome shows a big difference between human and avian strains of PB1-F2, and the authors suggested these differences are related to the PB1-F2 functions. I suggest that another full-length human or avian PB1-F2 could be added to the BioID2 and mess spec assay to prove that the difference between human and avian PB1-F2 was not random.
Reviewer #2 questions the specificity of the data generated in our work. It is true that this type of systematic analysis is prone to producing false positives. However, during our experiments, this aspect of the work was at the heart of our concerns and we implemented numerous controls to carry out our proximity marking experiments. For example, to validate our protocol, we performed proximity labelling of the viral protein NS1, of which many interactors are described. As an example, this allowed us to identify CPSF30, a well described interactor of NS1. Thus, by comparing our results with the previously described NS1 interactomes, we were able to validate our procedures. In addition, during our proximity labelling with PB1-F2, we integrated a GFP control to identify non-specific proteins. Lastly, the use of the "Crapome" application (Contaminant Repository for Affinity Purification; https://reprint-apms.org/?q=reprint-home) allowed us to impose a drastic filter to minimize the impact of contaminating proteins. Finally, we focused on 14-3-3 proteins that produced a consistent hit during the BioID2 screen and validated the interaction using conventional methods. Overall, we believe that the inclusion of additional PB1-F2 in this study would add confusion to the message delivered.
Minor issues:
- Line 63: “acid” should be “acids”.
Correction done
- Line 81: “Thus, and even if” should be “Thus, even if”.
Correction done
- Line 113: “during 45 min at room temperature” should be “for 45 min at room temperature”.
Correction done
- The protein alignment of PB1-F2 could be moved to Figure 1 from the supplementary.
Correction done

Reviewer 3 Report
Influenza A viruses express an accessory protein PB1-F2, but its function and effect on viral pathogenicity are controversial. In this study, the authors analyzed PB1-F2 protein interactomes of an avian H7N1 virus and a human H3N2 virus using a proximity-dependent biotin labeling approach. The data indicated common and differential pathways targeted by these PB1-F2 proteins, including pathways involved in the host response. The study is of interest for the study of possible role of PB1-F2 in influenza pathogenicity and host specificity. However, there are some issues, especially lack of key negative controls in some experiments, which need to be fixed before publication.
1. The amino acid difference between the avian and human viruses used in this study should be shown in the main text first (maybe in Fig 1), not in supplemental data.
2. Fig 2B: The authors should include control samples (without addition of exogenous biotin) to show the specificity and level of background of this Western blot analysis.
3. Fig 5C: The same experiment should be done with H7N1 PB1-F2 to confirm the specificity of this interaction.
4. Fig. 5D: A key control sample is missing. The authors need to determine the level of luciferase expression in a sample expressing both MDA5 and YWHAH without PB1-F2 to evaluate the effect of PB1-F2 on activation of IFN signaling.
5. Introduction (Lines 55-83): The rationale for this study is not justified well. The authors suggest that loss of expression of PB1-F2 is beneficial for the fitness of H1N1 isolates in humans and more generally in mammals (lines 63-65), although human H3N2 viruses maintain full length PB1-F2 (line 74).
6. Introduction (line 45-46): I guess the authors intend to state that ANP32 mediates the formation of dimers of polymerase complex required for genome replication.
7. Lines 96: It should be “Purification of recombinant PB1-F2 and fibrillation assays”.
Author Response
Reviewer #3
Comments and Suggestions for Authors
Influenza A viruses express an accessory protein PB1-F2, but its function and effect on viral pathogenicity are controversial. In this study, the authors analyzed PB1-F2 protein interactomes of an avian H7N1 virus and a human H3N2 virus using a proximity-dependent biotin labeling approach. The data indicated common and differential pathways targeted by these PB1-F2 proteins, including pathways involved in the host response. The study is of interest for the study of possible role of PB1-F2 in influenza pathogenicity and host specificity. However, there are some issues, especially lack of key negative controls in some experiments, which need to be fixed before publication.
We thank reviewer #3 for his constructive comments to improve our manuscript.
- The amino acid difference between the avian and human viruses used in this study should be shown in the main text first (maybe in Fig 1), not in supplemental data.
Figure 1 has been modified to include the sequence information of the 2 proteins studied.
- Fig 2B: The authors should include control samples (without addition of exogenous biotin) to show the specificity and level of background of this Western blot analysis.
Reviewer#3 make a good point. These important controls were made at the beginning of our study during the development. We verified that the anti-biotin Western blot did not reveal any signal in the absence of biotin addition. This was done and validated for both PB1-F2 as well as for the NS1 and PA-X proteins. Below is a blot illustrating these developmental experiments (see attached file). As the quality of these gel pictures is poor, we have chosen not to integrate them into the manuscript as recommended by reviewer #3, but we have modified the text to make it clear to the reader that these controls have been done. The following sentence has been added: “It should be noted that when biotin is not added to the medium, Western blots show no signal, demonstrating the specificity of the signal (data not shown).”
- Fig 5C: The same experiment should be done with H7N1 PB1-F2 to confirm the specificity of this interaction.
The control condition requested has been added as a supplemental information. The following sentence is now present in the result section:” The potential interaction between YWHAH and PB1-F2 from H7N1 was tested and no binding could be seen (Fig S1).”.
- Fig. 5D: A key control sample is missing. The authors need to determine the level of luciferase expression in a sample expressing both MDA5 and YWHAH without PB1-F2 to evaluate the effect of PB1-F2 on activation of IFN signaling.
The control condition requested by reviewer #3 has been added in Figure 5.
- Introduction (Lines 55-83): The rationale for this study is not justified well. The authors suggest that loss of expression of PB1-F2 is beneficial for the fitness of H1N1 isolates in humans and more generally in mammals (lines 63-65), although human H3N2 viruses maintain full length PB1-F2 (line 74).
This is correct, but we point out that, interestingly, H3N2 viruses that maintain PB1-F2 expression are generally more virulent than H1N1 viruses that have lost PB1-F2 expression since the 1940s. It should also be considered that H3N2 viruses are more recent pathogens in the human population compared to H1N1 viruses, which may explain the preservation of PB1-F2 expression in these viruses. Furthermore, the fact that the loss of PB1-F2 is beneficial for the adaptation of influenza in the human population is not only our suggestion but a hypothesis shared by other research groups: “Since the PB1-segment of the pandemic 1918, 1957 and 1968 strains originates from the avian reservoir, one could hypothesize that these pro-inflammatory effects of PB1-F2 are important for crossing the species barrier and get lost during the continuously adaptation process in the new host” Krumbholz et al. Medical Microbiology and Immunology (2011).
- Introduction (line 45-46): I guess the authors intend to state that ANP32 mediates the formation of dimers of polymerase complex required for genome replication.
This has been added to the introductory section.
- Lines 96: It should be “Purification of recombinant PB1-F2 and fibrillation assays”.
Correction done

Round 2
Reviewer 2 Report
The paper, " Comparison of PB1-F2 proximity interactomes reveals functional differences between human and avian influenza viruses." by Joëlle Mettier et al, describes the functional differences of PB1-F2 between H7N1 A/Turkey/Italy/977/1999 avian influenza strain and H3N2 A/Scotland/20/1974 human influenza strain. In this paper, BioID2 pull-down was performed, and the two PB1-F2 proteins share only a few interactors. When co-expression of PB1-F2 with YWHAH and MDA5, the H3N2 PB1-F2 exacerbated the IFN activity but the H7N1 PB1-F2 could not. In my opinion, the work is interesting and suitable for publishing in Viruses
Reviewer 3 Report
The authors responded to all previous comments and modified the article appropriately. I have no additional issues with this article.